# The SARS-CoV-2 Omicron BA.1 spike G446S mutation potentiates antiviral T-cell recognition

Chihiro Motozono [1,2] ✉, Mako Toyoda[1], Toong Seng Tan[1], Hiroshi Hamana [3], Yoshihiko Goto[1,4], Yoshiki Aritsu[5], Yusuke Miyashita[6,7], Hiroyuki Oshiumi [6], Kimitoshi Nakamura[7], Seiji Okada [8], Keiko Udaka[9], Mizuki Kitamatsu[5], Hiroyuki Kishi [3] & Takamasa Ueno [1] ✉

Although the Omicron variant of the SARS-CoV-2 virus shows resistance to neutralizing antibody, it retains susceptibility to the cellular immune response. Here we characterize vaccine-induced T cells specific for various SARS-CoV-2 variants and identified HLA-A*24:02-restricted CD8+ T cells that strongly suppress Omicron BA.1 replication in vitro. Mutagenesis analyses revealed that a G446S mutation, located just outside the N-terminus of the cognate epitope, augmented TCR recognition of this variant. In contrast, no enhanced suppression of replication is observed against cells infected with the prototype, Omicron BA.2, and Delta variants that express G446. The enhancing effect of the G446S mutation is lost when target cells are treated with inhibitors of tripeptidyl peptidase II, a protein that mediates antigen processing. These ex vivo analysis and in vitro results demonstrate that the G446S mutation in the Omicron BA.1 variant affects antigen processing/presentation and potentiates antiviral activity by vaccine-induced T cells, leading to enhanced T cell recognition towards emerging variants.

Current mRNA vaccines against the severe acute respiratory syndrome coronavirus 2 (SARS-CoV-2) employ the viral spike protein as the target antigen. These vaccines elicit neutralizing antibodies and T-cell responses to the spike protein that play a central role in defending against viral infection and viral replication in vivo. The SARS-CoV-2 Omicron variant (BA.1; B.1.1.529.1), first identified in November 2021, is a novel variant that has rapidly spread around the globe. BA.1 has >30 mutations in the spike protein[1] that contribute to reduced sensitivity to vaccine-induced antibody neutralization[2-4]. In contrast, vaccine-

induced SARS-CoV-2-specific T cells retain their reactivity to a number of different variants[5,6]. Recent studies demonstrated that T cells induced in vaccinated donors can cross-recognize the Omicron variant[6-10]. However, the characteristics of vaccine-induced T cells cross-reactive for the Omicron virus are only poorly understood.

T-cell epitopes are generated by the proteasome degradation of intracellular viral proteins into peptides that are subsequently trimmed by cytosolic aminopeptidases[11]. Some of these peptides are translocated via the transporter associated with antigen presentation

[1]Division of Infection and Immunity, Joint Research Center for Human Retrovirus Infection, Kumamoto University, Kumamoto 8600811, Japan. [2]Laboratory of Molecular Immunology, Immunology Frontier Research Center, Osaka University, Suita 5650871, Japan. [3]Department of Immunology, Faculty of Medicine, Academic Assembly, University of Toyama, Toyama 9300194, Japan. [4]Department of Respiratory Medicine, Faculty of Life Sciences, Kumamoto University, Kumamoto 8608556, Japan. [5]Department of Applied Chemistry, Faculty of Science and Engineering, Kindai University, Osaka 577-8502, Japan. [6]Department of Immunology, Graduate School of Medical Sciences, Faculty of Life Sciences, Kumamoto University, Kumamoto 8608556, Japan. [7]Department of Pediatrics, Graduate School of Medical Sciences, Kumamoto University, Kumamoto 8608556, Japan. [8]Division of Hematopoiesis, Joint Research Center for Human Retrovirus Infection, Kumamoto University, Kumamoto 8600811, Japan. [9]Department of Immunology, Kochi University, Kochi 783-8505, Japan. ✉e-mail: motozono@kumamoto-u.ac.jp; uenotaka@kumamoto-u.ac.jp

(TAP) into the endoplasmic reticulum (ER) lumen and loaded onto HLA class I molecules. Peptide/HLA class I complexes are released from the ER and transported via the Golgi to the plasma membrane, where they are presented for recognition by CD8+ T cells. Mutations within the peptide epitope directly affect HLA binding and T-cell recognition. In addition, mutations outside the epitope can affect T-cell recognition by interfering with the intracellular processing of virus-derived proteins[11,12]. For example, a mutation from alanine to proline at HIV-1 Gag residue 146, immediately preceding the NH2 terminus of a dominant HLA-B57-restricted epitope, prevented NH2-terminal trimming of the optimal epitope by the ER aminopeptidase I[12]. These studies suggest that certain mutations located outside and nearby immunodominant epitopes may affect the T-cell response to SARS-CoV-2 variants.

In this study, we demonstrated that a subset of vaccine-induced HLA-A*24:02-restricted T cells exhibited enhanced reactivity against the Omicron BA.1 variant. This enhanced reactivity was associated with a G446S mutation in the Omicron BA.1 spike protein that altered (enhanced) the processing and presentation of the associated antigenic peptide. Enhanced presentation of this epitope was also associated with greater inhibition of BA.1 replication by vaccine-induced T cells.

## Results

### Vaccine-induced immunodominant responses in the context of HLA-A*24:02

HLA-A*24:02 is one of the most widely distributed HLA-I alleles globally (predominantly in East Asia)[13]. A bioinformatic study identified three candidate HLA-A*24:02-binding peptides in the spike protein, RFDNPVLPF (RF9; residues 78–86), NYNYLYRLF (NF9; residues 448–456), and QYIKWPWYI (QI9; residues 1208–1216)[14]. All three peptides bound tightly to HLA-A*24:02 (Fig. 1a). To investigate vaccine-induced immunodominant responses in the context of HLA-A*24:02, we obtained PBMCs from individuals that had been vaccinated with two doses of BNT162b2 or mRNA-1273 vaccines (n = 30) (Supplementary Table 1). PBMC were stained with peptide/HLA tetramers of the three epitopes RF9/HLA-A*24:02 (RF9/A24), NF9/HLA-A*24:02 (NF9/A24), and QI9/HLA-A*24:02 (QI9/A24) (Fig. 1b and Supplementary Fig. 1a). NF9/A24- and QI9/A24-specific T cells were detected in 11 (36.7%) and 14 (46.7%), respectively, of 30 HLA-A*24:02+ vaccinated donors (Fig. 1c). RF9/A24-specific T cells were detected in only 2 of 30 individuals (6.7%). These data indicate that CD8+ T cells specific for NF9/A24 and QI9/A24 are predominantly induced in HLA-A*24:02+ vaccinated donors.

To analyze the recognition of SARS-CoV-2 variants by NF9/A24- and QI9/A24-specific T cells, we stimulated PBMCs from 10 donors (GV9, 15, 16, 26, 31, 33, 34, 36, 60, and 61) with the NF9 or QI9 peptides. After 14 days, proliferating T cells were evaluated for the upregulation of two activation markers, CD25 and CD137 (Fig. 1d and Supplementary Fig. 1b), as previously described[13,15]. The percentages of CD25+CD137+ T cells after stimulation with the NF9 peptide (median 3.5%) and QI9 peptide (median 4.5%) were significantly higher than those in the presence of the HIV irrelevant peptide (median 0.5%) (Fig. 1e, p = 0.0020 by Wilcoxon matched-pairs signed-rank test; versus the irrelevant peptide). There was no significant difference in the frequencies of NF9/A24 and QI9/A24-specific T cells (Fig. 1e, p = 0.9219 by Wilcoxon matched-pairs signed-rank test), consistent with the tetramer staining data (Fig. 1c). Taken together, our ex vivo and in vitro data indicate that NF9/A24 and QI9/A24 are immunodominant epitopes presented by HLA-A*24:02 in BNT162b2 or mRNA-1273 vaccinated donors.

### Vaccine-induced NF9/A24-specific T cells efficiently recognize target cells expressing the Omicron BA.1 spike protein

To analyze the recognition of the prototype (D614G-bearing B.1 lineage), Omicron BA.1, and Delta (B.1.617.2) variants by vaccine-induced

HLA-A*24:02-restricted T cells, we established T-cell lines by stimulating of PBMCs from two HLA-A*24:02+ vaccinated donors (GV34 and GV60) with the NF9 or QI9 peptides (Supplementary Fig. 1c–e). The resulting T-cell lines were tested for the recognition of A549-ACE2-A2402 cells (Supplementary Fig. 1f) that had been engineered to express spike protein from the prototype, Omicron BA.1, or Delta variants. These target cell lines expressed comparable levels of spike protein, as determined by western blot (Supplementary Fig. 1g). Interestingly, the level of IFN-γ production by the NF9-specific T-cell lines from both donors was higher toward target cells expressing Omicron BA.1 spike protein and lower toward cells expressing Delta spike protein, compared to cells expressing the prototype spike protein (Fig. 2a). On the other hand, the level of IFN-γ production by the QI9-specific T-cell lines was comparable against target cells expressing Omicron BA.1, Delta, and prototype spike proteins (Fig. 2b). The enhanced sensitivity of NF9/A24-specific T cells to Omicron BA.1 spike protein, compared to QI9/A24-specific T cells, was maintained at different E:T ratios (Fig. 2c). Analysis of a further six donors confirmed these results (Fig. 2d, p = 0.0002 by Mann–Whitney test versus prototype). Of note, diminished T-cell sensitivity against target cells expressing the Delta spike protein is due to the presence of an L452R mutation in the NF9 peptide, a hallmark of the Delta spike protein (Fig. 3a). These data extend our previous analysis of T-cell recognition of target cells pulsed with the mutant peptide by T cells from COVID-19 convalescents[13] and vaccinated donors[15]. In contrast, there was no mutation within the NF9 epitope in Omicron BA.1 (Fig. 3a), suggesting that Omicron BA.1 mutation of amino-acid residues outside the NF9 peptide affects the sensitivity of the target cells to NF9/A24-specific T cells.

### The G446S mutation in the SARS-CoV-2 Omicron BA.1 spike protein is responsible for enhanced NF9/A24-specific TCR recognition

We next identified TCR pairs specific for the NF9/A24 and QI9/A24 epitopes by single-cell sorting with NF9/A24 and QI9/A24 tetramers[16]. These studies focused on TCR sequences from 4 donors (Supplementary Table 2). Pairs of TCR α and β chains were reconstituted in a TCR-deficient NFAT-luciferase reporter cell line. TCRs specific for NF9/A24 (#5-3 and #12-3) and QI9/A24 (#43 and #57) were expressed on the cell surface and bound cognate tetramers at levels comparable to those of T-cell lines (Fig. 3b). Both of the NF9/A24-specific TCR lines responded to cells expressing the Omicron BA.1 spike protein to a greater extent than those expressing the prototype, Alpha (B.1.1.7), Beta (B.1.351), and Gamma (P.1) spike proteins (Fig. 3c). These data indicate that the level of the NF9 peptide on the cell surface was enhanced in target cells expressing the Omicron BA.1 spike protein. Consistent with this, peptide-titration experiments using the #5-3 TCR indicated that the amount of NF9 peptide expressed on the target cells was almost threefold greater on cells expressing Omicron BA.1 spike protein than cells expressing the prototype spike protein (Supplementary Fig. 2a). Also, the TCR responses were significantly reduced in cells expressing Delta and Lambda (C.37 lineage) spike protein, agreeing with our previous reports that L452R and L452Q in Delta and Lambda, respectively, mediate escape from vaccine-induced NF9/A24-specific T cells[15]. In contrast, both QI9/A24-specific TCRs responded comparably to cells expressing the spike protein from the prototype and variants (Fig. 3c).

It has been shown that mutations adjacent to T-cell epitopes in HIV-1 can affect antigen processing and subsequent display to T cells[12,17–19]. To determine whether this is also true for SARS-CoV-2 -specific T cells, we introduced mutations adjacent to the NF9 peptide sequence in BA.1 spike protein[1]. There are two mutations, N440K and G446S, located at 8 and 2 amino acids preceding the NF9 peptide sequence in Omicron BA.1 spike protein (Fig. 3a). Therefore, we sought to examine whether N440K or G446S could enhance epitope recognition by NF9/A24-specific T cells.

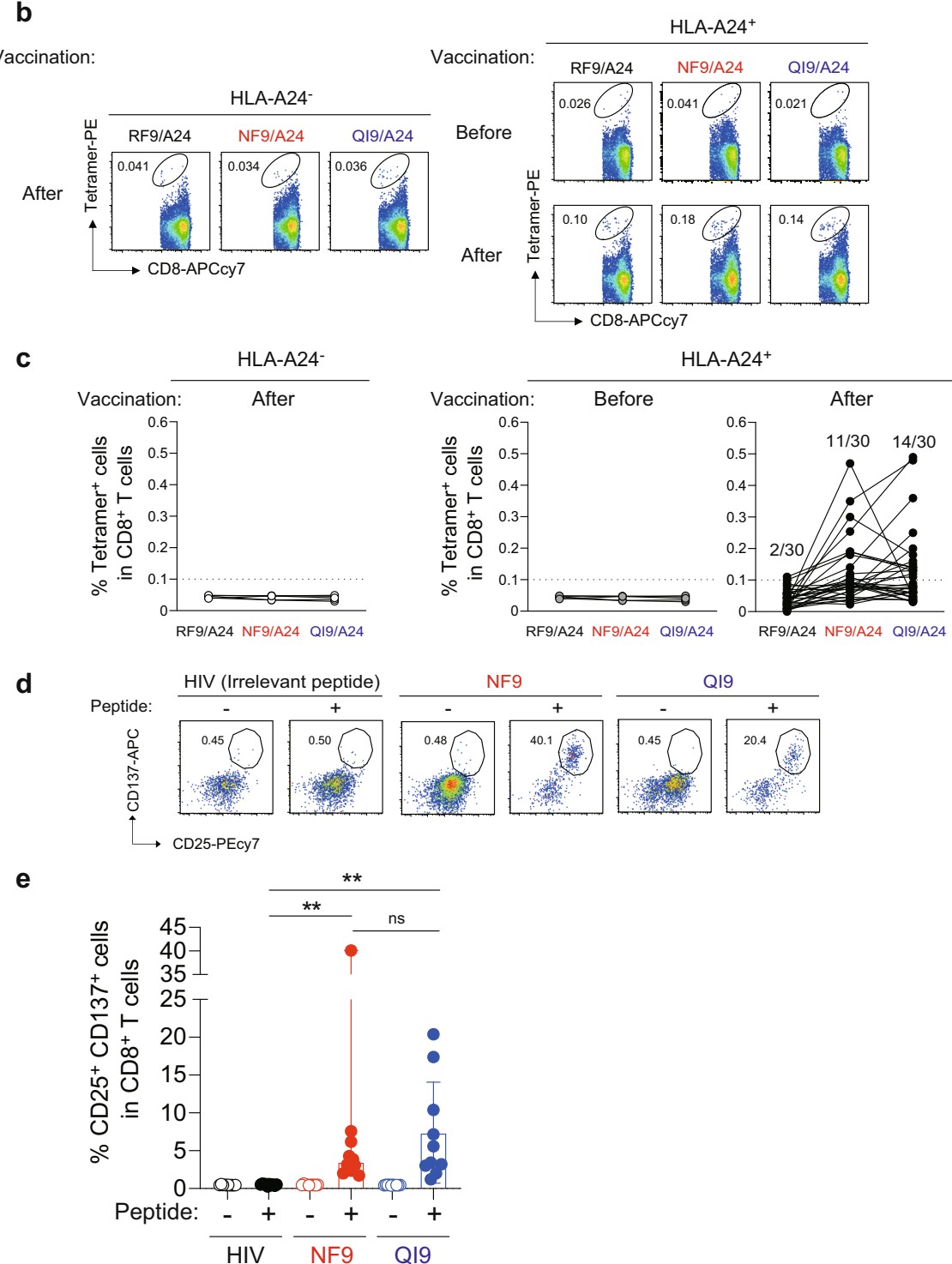

The introduction of both mutations, or the G446S mutation alone, to the prototype spike protein, resulted in significantly enhanced recognition by NF9/A24-specific TCRs. In contrast, the introduction of N440K alone and reversion of the sequence (S446G) in Omicron BA.1 spike protein was recognized by NF9/A24-specific TCRs at levels similar to those induced by the prototype spike protein (Fig. 3d).

Moreover, QI9/A24-specific TCRs comparably recognized all target cells tested, consistent with a comparable level of spike protein expression, as determined by western blot (Supplementary Fig. 2b). Taken together, our findings indicate that the N-terminal adjacent G446S mutation of the NF9 epitope in Omicron BA.1 spike protein is responsible for enhancing NF9/A24-specific TCR recognition.

**Fig. 1 | Detection of HLA-A*24:02-restricted antigen-specific T cells in vaccinated donors. a** Binding of spike-derived epitopes to HLA-A*24:02 by HLA stabilization assay. **b, c** Detection of HLA-A*24:02-restricted antigen-specific T cells in PBMCs by HLA tetramers. **b** Representative FACS plots showing tetramer⁺ CD8⁺ T cells of an HLA-A24-negative vaccinated donor, GV17, and an HLA-A24-positive donor, GV16, before (upper) and after vaccination (lower). See also Supplementary Fig. 1a (Gating strategy). **c** The median of the percentage of RF9/A24, NF9/A24, and QI9/A24 tetramer⁺CD8⁺ T cells. The median percentages of T cells in vaccinated HLA-A*24:02-positive donors ($n = 30$). The median percentages of T cells in vaccinated HLA-A*24:02-negative donors ($n = 5$; GV4, GV12, GV17, GV25, and GV27) and non-vaccinated/seronegative HLA-A*24:02-positive donors ($n = 5$; GV16, GV32, GV33, GV34, and GV36) are shown as negative controls. **d, e** Detection of HLA-A*24:02-restricted antigen-specific T-cell lines in vitro. **d** Representative FACS plots showing surface expression of CD25 and CD137 on the CD8⁺ T-cell subset of a

vaccinated donor, GV60. See also Supplementary Fig. 1b (Gating strategy). **e** The median of the percentage of CD25⁺CD137⁺ cells after stimulation with the NF9 peptide (median 3.5%) and QI9 peptide (median 4.5%) and HIV irrelevant peptide (0.5%) among CD8⁺ T cells in vaccinated HLA-A*24:02-positive donors ($n = 10$). See also Supplementary Table 1 (donor information). **b, d** the numbers in the FACS plot represent the percentage of gated cells among CD8⁺ T cells. In **c**, the median of % tetramer⁺ population in HLA-A24 negative donors was <0.05%. HLA-A24⁺ donors showing a >0.1% tetramer⁺ response were considered to be responders (>0.1%, the median plus 4× SD values of the negative controls). **e** Data are expressed as median and a statistically significant difference between the indicated peptides (**$p = 0.0020$) is determined by two-tailed Wilcoxon matched-pairs signed-rank test. ns, no statistical significance ($p = 0.92$). Source data are provided as a Source Data file.

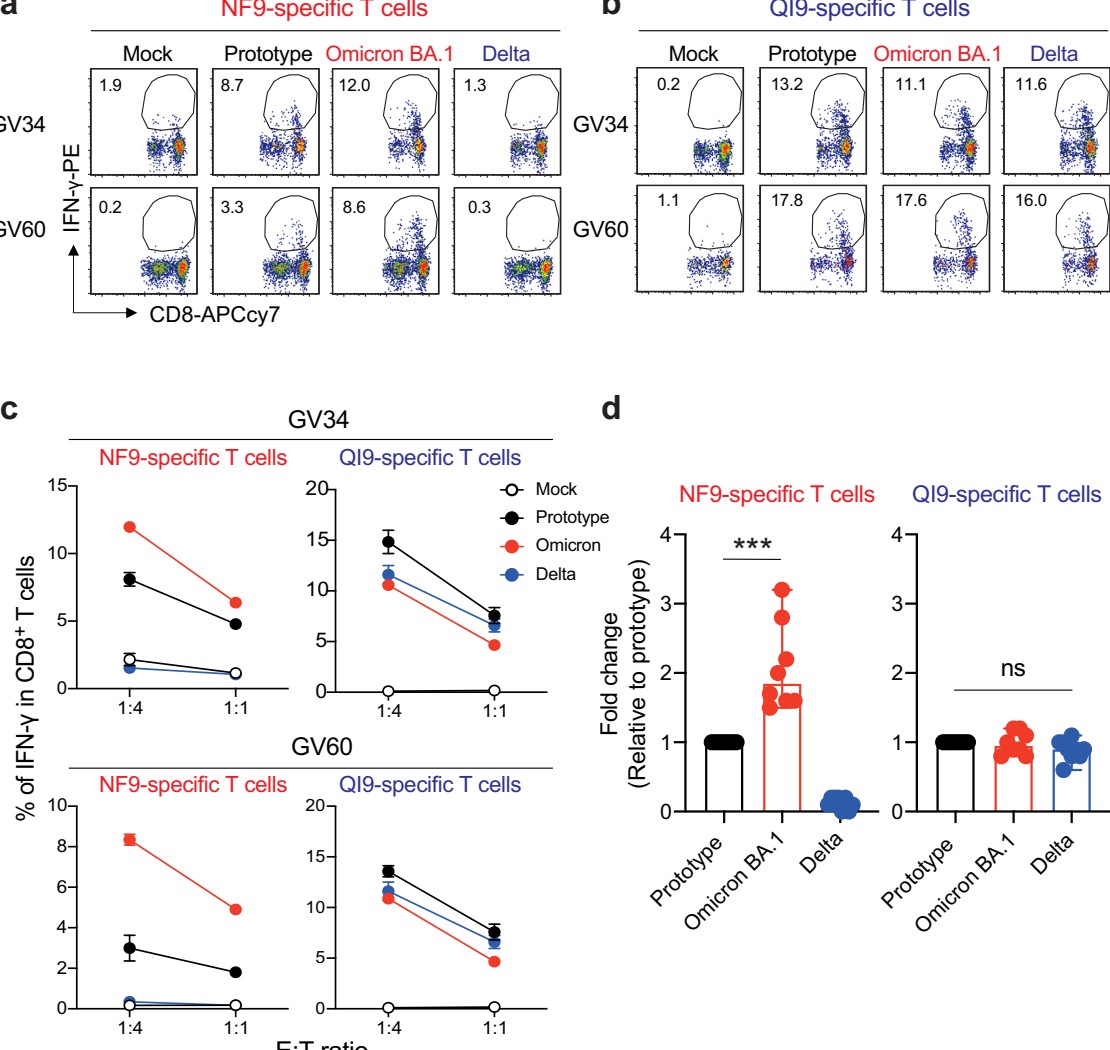

**Fig. 2 | T-cell recognition to target cells expressing variant spike protein.**
**a, b** HLA-A24-positive T-cell lines of vaccinated donors were stimulated with A549-ACE/A2402 (Supplementary Fig. 1f) expressing spike protein derived from prototype, Omicron, and Delta variants. Representative FACS plots showing intracellular expression of IFN-γ in the NF9-specific CD8⁺ T cells (**a**) and QI9-specific CD8⁺ T cells (**b**) in two vaccinated donors (GV34 and GV60). **c** The level of IFN-γ production of NF9- and QI9-specific T cells in response to spike-expressing target cells in two vaccinated donors, GV34 (upper) and GV60 (lower). **d** Fold changes in IFN-γ

expression by NF9-specific T cells (left) and QI9-specific T cells (right) compared to the target cells expressing prototype spike in eight vaccinated donors (GV15, 26, 31, 33, 34, 36, 60, and 61) are shown. **a, b** The numbers in the FACS plot represent the percentage of IFN-γ⁺ cells among CD8⁺ T cells. **c** The assay was performed in triplicate, and the means are shown with the SD. **d** The assay was performed in triplicate, and the median is shown. A statistically significant difference versus prototype spike (***$p = 0.0002$) is determined by a two-tailed Mann–Whitney test. ns, no statistical significance. Source data are provided as a Source Data file.

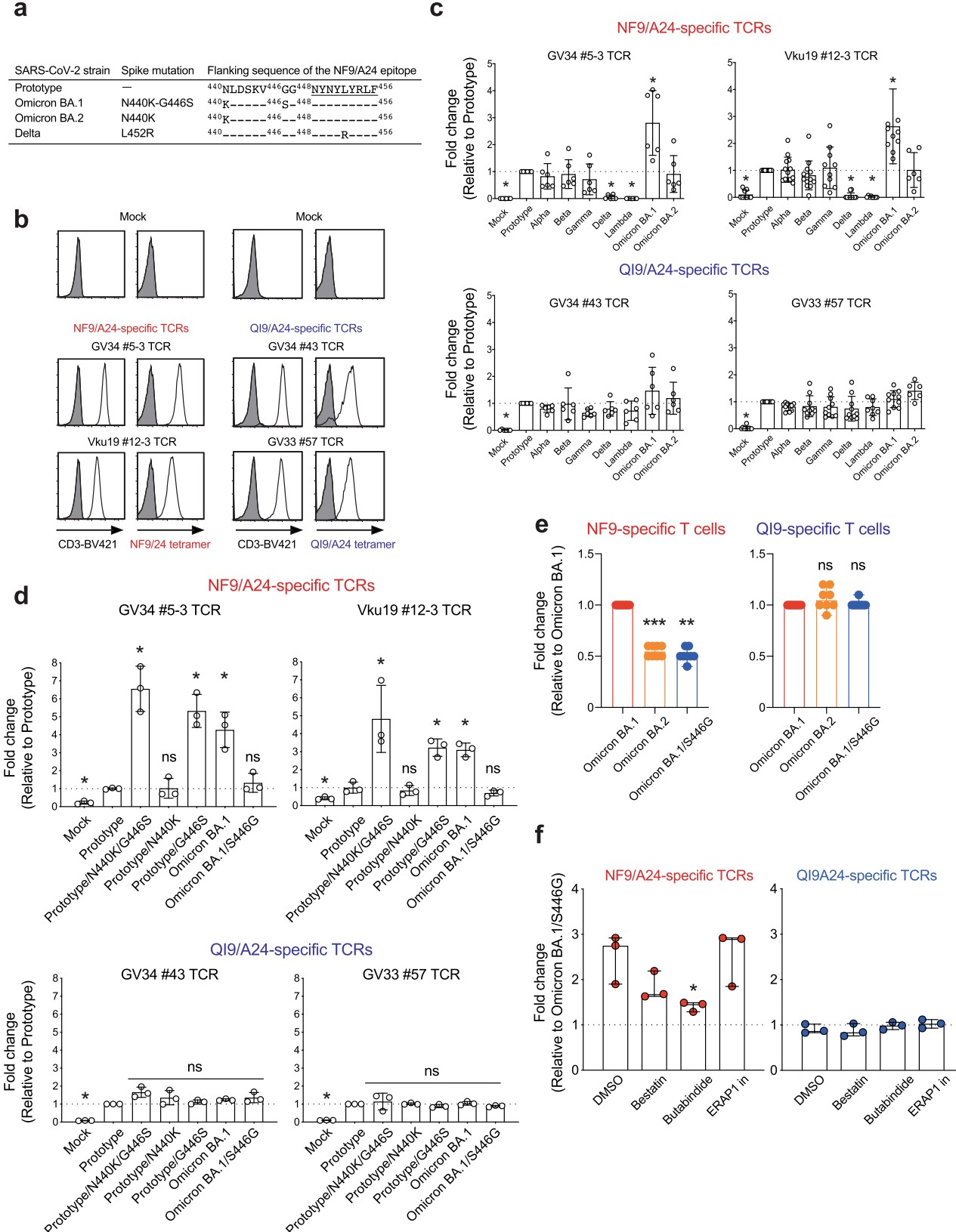

**Enhanced T-cell recognition is diminished against Omicron BA.2 spike protein due to the lack of the G446S mutation**

The SARS-CoV-2 Omicron variant (B.1.1529) is comprised of three major lineages BA.1 (B.1.1529.1), BA.2 (B.1.1529.2), and BA.3 (B.1.1529.3)[1]. Following the initial spread of BA.1, BA.2 is now becoming the prevalent variant as of April 2022. Because the G446S mutation is lost in

Omicron BA.2, we investigated the recognition of Omicron BA.2 by NF9-specific T cells. BA.2 spike protein-expressing cells were recognized by NF9/A24-specific TCRs to a lesser extent than those expressing BA.1 spike protein, and comparable to those expressing the prototype (Fig. 3c). Furthermore, NF9-specific T cells from eight vaccinated HLA-A*24:02+ donors exhibited decreased levels of IFN-γ in

**Fig. 3 | Identification of mutation associated with increased TCR sensitivity.**
**a** Spike-derived HLA-A24-restricted NF9 epitopes and the N-terminal flanking region from the variant. **b**–**d** TCR-peptide/HLA interaction on a TCR-transduced Jurkat NFAT-luciferase reporter cell. **b** Jurkat cells alone (shaded histogram) or those expressing NF9/A24-specific TCRs (#5-3 and #12-3) or QI9/A24-specific TCRs (#43 and #57) (open histogram) were stained with anti-CD3 mAb and their cognate HLA-A24 tetramers and then analyzed by flow cytometry. **c, d** The level of peptide/HLA complexes was evaluated by NFAT-luciferase reporter activity of TCR-transduced Jurkat cells. Data of NF9/A24-specific TCRs and QI9/A24-specific TCRs are shown. Fold changes of reporter activity by NF9/A24- and QI9/A24-specific TCRs compared to the target cells expressing prototype spike protein are shown. **e** Fold changes of IFN-γ by NF9-specific T cells (left) and QI9-specific T cells (right) compared to the target cells expressing Omicron BA.1 spike protein in eight vaccinated donors (GV15, 26, 31, 33, 34, 36, 60, and 61) are shown. The assay was performed in triplicate, and data are expressed as median. A statistically significant difference versus Omicron BA.1 spike was determined by a two-tailed Mann–Whitney test. $p = 0.0002$ (***$p < 0.001$), and $p = 0.0078$ (**$p < 0.01$) versus Omicron BA.2 and Omicron BA.1/S446G, respectively. ns, no statistical significance. **f** The level of peptide/HLA complexes is evaluated following treatment with inhibitors, bestatin (120 μM), butabindide (150 μM), and ERAP1 inhibitor (50 μM). The effect of inhibitors was evaluated as reporter activity and shown by fold change of

the level of peptide/HLA on target cells expressing Omicron BA.1 reversion S446G spike protein. Data are expressed as a median. Statistical analysis versus DMSO alone was determined by one-way ANOVA with multiple comparisons by Bonferroni correction. $p = 0.0456$ (*$p < 0.05$) versus butabindide. **c** A statistically significant difference versus prototype was determined by an unpaired two-tailed Student's $t$ test (*$p < 0.05$) and $p$ values are shown as the followings: Mock ($p < 0.0001$), Delta ($p < 0.0001$), and Lambda ($p < 0.0001$) in NF9-specific TCRs and Omicron BA.1 in GV34 #5-3 TCR ($p = 0.0043$) and Vku #12-3 TCR ($p = 0.003$), and Mock ($p < 0.0001$) in QI9-specific TCRs (GV34 #43 and GV33 #57 TCR). ns, no statistical significance. Data are expressed as mean ± SD. **d** A statistically significant difference versus prototype was determined by an unpaired two-tailed Student's $t$ test (*$p < 0.05$) and p values are shown as the followings: Mock ($p = 0.0001$), Prototype/N440K/G446S ($p = 0.0016$), Prototype/G446S ($p = 0.0012$) and Omicron BA.1 (0.0046) in GV34 #5-3 TCR and Mock ($p = 0.0321$), Prototype/N440K/G446S ($p = 0.0274$), Prototype/G446S ($p = 0.0025$) and Omicron BA.1 ($p = 0.0017$) in Vku19 #12-3 TCR, and Mock ($p < 0.0001$) in QI9-specific TCRs (GV34 #43 and GV33 #57 TCR). ns, no statistical significance. Data are expressed as mean ± SD. **c, d, f** The assay was performed in triplicate or quadruplicate, and data are representative of two or three independent experiments. Source data are provided as a Source Data file.

response to stimulation with target cells expressing Omicron BA.2 spike protein, similar to the levels elicited by the Omicron BA.1 reversion S446G (Fig. 3e, $p = 0.0002$ and $p = 0.0078$ by Mann–Whitney test versus Omicron BA.2, Omicron BA.1/S446G, respectively). In contrast, QI9-specific T cells from the same donors produced comparable levels of IFN-γ in response to stimulation with target cells expressing all spike proteins tested (Fig. 3e). Together, these data suggest that the introduction of a serine at position 446 of Omicron BA.1 spike protein is sufficient to induce enhanced T-cell recognition of the NF9 epitope. This enhanced recognition is diminished against Omicron BA.2 spike protein due to the absence of the G446S mutation.

**Tripeptidyl peptidase II (TPPII) inhibitor reduces the enhanced recognition of the NF9 epitope**

To determine how the G446S mutation affects the antigen processing pathway, we first performed a TCR-sensitivity assay using Omicron BA.1 spike protein-expressing target cells pre-treated with MG-132, an inhibitor for proteasomes. There was no statistical difference between NF9/A24 and QI9/A24-specific TCRs in the presence of MG-132 (Supplementary Fig. 2c, $p > 0.8$ by unpaired two-tailed Student's $t$ test). Next, we performed the assay with a panel of protease inhibitors that are involved in N-terminal processing/trimming of the peptide; e.g., bestatin (aminopeptidase inhibitor), butabindide (tripeptidyl peptidase II inhibitor) and ERAP1 inhibitor compound 3[11]. The enhanced sensitivity of NF9/A24-specific TCRs (GV34 #5-3, Vku19 #12-3, and GV34 #2-2) to Omicron BA.1 spike protein was significantly reduced in the presence of TPPII inhibitor (Fig. 3f, $p = 0.0456$ by ANOVA, with multiple comparisons by Bonferroni correction; versus DMSO alone). The enhanced sensitivity was modestly reduced in the presence of bestatin, but in this case, the difference did not reach statistical significance (Fig. 3f, $p = 0.27$). In contrast, there was no difference in the sensitivity of QI9/A24-specific TCRs (GV34 #43, GV33 #57, and GV36 #10-2). These data suggest that TPPII might be one of the proteases involved in the efficient generation of the NF9 epitope.

**NF9-specific T cells efficiently suppress the replication of Omicron BA.1, but not the BA.2, viral variants**

We next investigated whether NF9-specific T cells have a superior capacity to suppress the replication of the Omicron BA.1 viral variant. A549 cells expressing ACE2/A2402 were infected with SARS-CoV-2 viral variants and cocultured with T-cell lines specific for NF9/A24 and QI9/A24. Suppression of viral replication by T cells is evaluated by the amount of viral RNA in the supernatant at 72 h after infection. NF9/A24 and QI9/A24-specific T cells isolated from donor GV36, significantly

inhibited replication of the prototype virus at 72 h (Fig. 4a, $p = 0.009$ and $p = 0.0036$ versus without T cells by unpaired two-tailed Student's $t$ test). NF9-specific T cells isolated from donors GV26, GV36, and GV60 also suppressed replication of Omicron BA.1 variant replication to a greater extent than the prototype or Omicron BA.2 (Fig. 4b). No suppression was observed against the Delta variant (Fig. 4b), presumably due to the T-cell escape mutation L452R in the Delta spike protein (Fig. 2c, d). On the other hand, QI9-specific T cells comparably suppressed viral replication of prototype, and these variants in all three donors tested (Fig. 4c). We also tested the antiviral activity of T cells from three additional donors and confirmed that NF9-specific T cells have the capacity to inhibit Omicron BA.1, but not BA.2 replication to a greater extent than the prototype. These data indicate that vaccine-induced T cells can have an enhanced capacity to recognize and suppress emerging SARS-CoV-2 BA.1 variant.

## Discussion

In this study, we report that vaccine-induced HLA-A*24:02-restricted, NF9-specific T cells efficiently recognize target cells expressing the Omicron BA.1 spike protein and strongly suppress viral replication of the Omicron BA.1 variant compared to the prototype virus. The G446S mutation in Omicron BA.1 spike protein, located adjacent to the N terminus of the NF9 epitope (residues 448–456), is responsible for the efficient generation of the epitope. This is presumably due to enhanced antigen processing and presentation of the epitope. These data indicate that vaccine-induced T cells can have an enhanced capacity to cross-recognize and suppress emerging SARS-CoV-2 variants.

The generation of HLA class I-restricted peptides is profoundly influenced by amino acid variations, not only within, but also around the core epitope. Changes in the epitope-flanking region can result in the inhibition of epitope presentation or a significant increase in the generation of the epitope[19]. In our study, we used a TCR-based quantification assay to demonstrate that the presentation of the NF9 epitope on the surface of Omicron BA.1 spike protein-expressing cells was estimated to be almost threefold increased relative to that of the prototype. The finding that T-cell recognition of NF9 epitope was reduced when the Omicron BA.1 spike protein-expressing target cells were pre-treated with butabindide, an inhibitor of TPPII, suggested that generation of NF9 epitope requires TPPII-mediated removal of 2-3 amino acids from the N terminus of the peptide as TPPII is known to mediate this process[20]. This finding is consistent with reports that the flanking regions of some HIV-1 epitopes impact proteasomal processing of the epitope[12,17]. Further studies are needed to clarify how mutations in the spike protein and other proteins in SARS-CoV-2

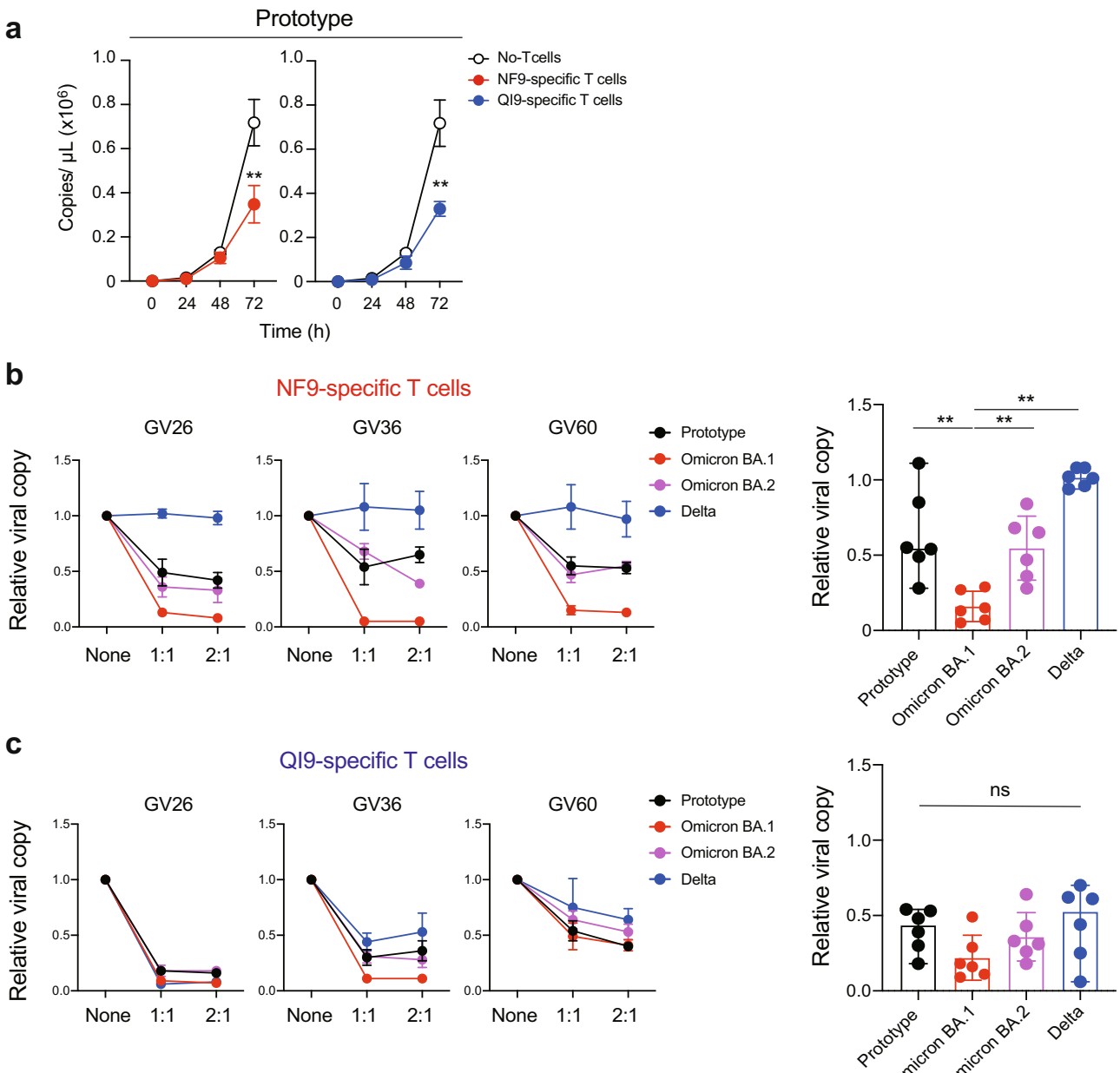

**Fig. 4 | Inhibition of SARS-CoV-2 viral replication by HLA-A24-restricted T-cell lines from vaccinated donors. a** Viral replication of the prototype in the presence of NF9- or QI9-specific T cells from GV36, or without T cells. **b, c** Inhibition of viral replication by NF9-specific T cells (**b**) and QI9-specific T cells (**c**) compared to without T cells in three vaccinated donors are shown. **a** A statistically significant difference versus without T cells (**$p < 0.01$) was determined by an unpaired two-tailed Student's $t$ test ($p = 0.009$ and $p = 0.0036$ versus without NF9 and QI9-specific T cells, respectively. Data are expressed as mean ± SD. **b, c** A statistically significant difference versus Omicron BA.1 (**$p < 0.01$) was determined by a two-tailed Mann–Whitney test and $p = 0.0043$, 0.0043, and 0.002 versus prototype, Omicron BA.2 and Delta, respectively. ns, no statistical significance. Data are expressed as median. **a–c** Assay was performed in triplicate or quadruplicate and data are representative of two independent experiments. Source data are provided as a Source Data file.

variants affect antigen processing/presentation for T-cell recognition, providing better insights for the rational design of vaccine antigens to induce efficient cellular immunity.

We and others previously reported that the NF9 is an immunodominant epitope presented by HLA-A*24:02 both in convalescent[13,21–23] and vaccinated donors[15]. However, the L452R and L452Q mutations in Delta/Epsilon and Lambda variants, respectively, conferred escape from NF9-specific T-cell responses[13,15]. In contrast, in this study, we demonstrated that NF9-specific T cells efficiently recognized target cells expressing Omicron BA.1 spike protein and suppressed viral replication of the Omicron BA.1 variant to a greater extent than that of the prototype. Interestingly, however, NF9-specific T cells only recognized and

suppressed viral replication of the closely related variant Omicron BA.2 comparable to the prototype. This is presumably due to the absence of G446S mutation in BA.2 and prototype. It will be interesting to see whether vaccine-induced T cells respond comparably, or differently, or control replication of SARS-CoV-2 variants of concern in the context of HLA-A*24:02 in vaccinated donors.

IFN-γ ELISpot or AIM (Activation-Induced marker) assays using overlapping peptides are powerful methods to evaluate the breadth of T-cell responses to overall viral proteins in vaccinated and COVID-19 convalescent donors[24,25]. Recent studies using these assays have demonstrated that T cells in vaccinated donors and convalescents can cross-recognize Omicron variants[6–10]. However, these assays do not

reveal antiviral functions of individual T cells against variants of concern, including the Omicron variant and the effect of mutations on antigen processing/presentation in virus-infected cells. Here, we found that a mutation located outside the epitope could enhance the antiviral activity of vaccine-induced T cells against the Omicron BA.1 variant. Thus, the antiviral assay for T cells demonstrated in this study will be tremendously useful for future vaccine development, and the combination of this quantitative assay with qualitative assays by ELISpot assays would be important to access vaccine efficacy against variants.

## Methods

### Ethics statement
For the use of human specimens, all protocols involving human subjects recruited at Kumamoto University were reviewed and approved by the Institutional Review Boards of Kumamoto University (approval numbers 2074 and 477). All human subjects provided written informed consent.

### Collection of human PBMCs
Human PBMCs were obtained from thirty HLA-A*24:02-positive BNT162b2 or mRNA-1273 vaccinated donors (median age: 24, range: 18–79, 67% male), five HLA-A*24:02-negative BNT162b2-vaccinated donors (median age: 24, range: 18–28, 60% Female) (Supplementary Table 1). PBMCs were purified by a density gradient centrifugation using Ficoll-Paque Plus (GE Healthcare Life Sciences, Cat# 17-1440-03) and stored in liquid nitrogen until further use.

### Cell culture
A549-ACE2/A2402 cells, the A549 cells stably expressing human ACE2 and HLA-A*24:02-IRES-GFP, were generated by retroviral transduction as previously described[13,16] and were maintained in Ham's-F12 (Wako, Cat# 080-08565) containing 10% fetal bovine serum (FBS). C1R cells expressing HLA-A*24:02 (C1R-A2402)[26] and TCR-deficient Jurkat cells expressing luciferase gene (JurkatΔ-Luc) were maintained in RPMI 1640 medium (Thermo Fisher Scientific, Cat# 11875101) containing 10% FBS.

### Virus
Four clinically isolated SARS-CoV-2 lineages were used: SARS-CoV-2 Wuhan strain [SARS-CoV-2/Hu/DP/Kng/19-020 (DDBJ Accession ID: LC528232)], was provided from Kanagawa Prefectural Institute of Public Health. A B.1.617.2 (Delta) lineage [hCoV-19/Japan/TKYK01734/2021 (GISAID Accession ID: EPI_ISL_2080609)], and a B.1.1.529 (Omicron/BA.1) lineage [hCoV hCoV-19/Japan/TKYX00012/2021 (GISAID Accession ID: EPI_ISL_8559478)] were provided from Tokyo Metropolitan Institute of Public Health, Tokyo, Japan. B.1.1.529 (Omicron/BA.2) lineage [hCoV hCoV-19/Japan/TY40-385-P1/2022 (GISAID Accession ID: EPI_ISL_9595859)] were provided from National Institute of Infectious Diseases, Tokyo, Japan.

### The peptide-dependent stabilization assay
The HLA binding of peptide was analyzed as previously described[27]. Briefly, TAP-deficient C1R-A24 cells were incubated at 26 °C overnight. A total of $1 \times 10^5$ cells was incubated with 1 μM β2-microglobulin (β2m) and graded concentrations of peptides in 96-well U-bottom plates. Cells were incubated at 26 °C for 1 h and then at 37 °C for 4 h. At the end of the incubation, unbound peptides were removed, and cells were stained with FITC-labeled HLA-A24-reactive mAb 17A12 (20 μg/ml) provided by Dr. Ulrich Hämmerling and analyzed by FACScan (BD, San Jose, CA, USA). The log Kd values for 50% binding were calculated from the mean fluorescence intensity (MFI) with Cell QuestTM. The log Kd values were normalized by including high-binder peptide (TYLPTNASL) and low-binder peptide (RVWESATPL) in each experiment.

### Tetramer staining
SARS-CoV-2-derived peptides-loaded MHC class I tetramers were generated by QuickSwitch™ Quant HLA-A*24:02 Tetramer Kit-PE (MBL International Corporation, Cat# TB-7302-K1) according to the manufacturer's protocol. The rate of peptide exchange was quantitated by flow cytometry, and the tetramers, at a rate of more than 90%, were used for staining PBMCs. After treatment with a protein kinase inhibitor, Dasatinib (Cat# A10290-25, AdooQ, 50 nM) for 30 min at 37 °C, PBMCs were stained with tetramers for 30 min on ice. After tetramer staining, cells were counterstained with anti-PE unconjugated mAb (PE001, Biolegend, 1/10 dilution) for 20 min on ice and surface stained with the following antibodies: CD3 BV421 (UCHT1, 1/50 dilution), CD8 APCcy7 (RPA-T8, 1/100 dilution), CD14 PerCP/Cy5.5 (HCD14, 1/100 dilution), CD19 PerCP/Cy5.5 (HIB19, 1/100 dilution; Biolegend) was performed. Dead cells were stained with 7-aminoactinomycin D (Biolegend, Cat# 420404). After incubation for 20 min on ice, the cells were fixed with 1% paraformaldehyde (Nacalai Tesque, Cat# 09154-85), and the levels of tetramer⁺CD8⁺ T cells were analyzed by flow cytometry using a FACS Canto II (BD Biosciences). The data obtained by flow cytometry were analyzed with FACS Diva v9.0 (BD) and FlowJo software v10 (Tree Star).

### Activation-induced marker assay
An activation-induced marker assay was performed as previously described[13]. Briefly, human PBMCs were pulsed with 100 nM of the NF9 peptide (NYNYLYRLF, residues 448-456 of the SARS-CoV-2 spike protein) and the QI9 peptide (QYIKWPWYI, residues 1208-1216 of the SARS-CoV-2 spike protein) maintained in RPMI 1640 medium (Thermo Fisher Scientific, Cat# 11875101) containing 10% FBS and 30 U/ml recombinant human IL-2 (Peprotec, Cat# 200-02) for 14 days. The HLA-A*24:02-restricted RF10 peptide (RYPLTFGWCF, residues 134-143 of the HIV Nef protein) at a concentration of 100 nM was included as a negative control. The in vitro expanded CD8⁺ T cells (i.e., T-cell lines) were restimulated with or without the peptide. After incubation at 37 °C for 24 h, the cells were washed, and surface stained with following antibodies: CD3 FITC (UCHT1, 1/100 dilution), CD8 APCcy7 (RPA-T8, 1/100 dilution), CD14 PerCP/Cy5.5 (HCD14, 1/100 dilution), CD19 PerCP/Cy5.5 (HIB19, 1/100 dilution), CD25 PEcy7 (M-A251, 1/50 dilution) and CD137 APC (4B4-1, 1/50 dilution; Biolegend). Dead cells were stained with 7-aminoactinomycin D (Biolegend, Cat# 420404). After incubation for 20 min on ice, the cells were fixed with 1% paraformaldehyde (Nacalai Tesque, Cat# 09154-85), and the levels of protein surface expression were analyzed by flow cytometry using a FACS Canto II (BD Biosciences). The data obtained by flow cytometry were analyzed with FACS Diva v9.0 (BD) and FlowJo software v10 (Tree Star).

### Plasmid construction
Plasmids expressing the SARS-CoV-2 spike proteins of the parental (D614G-bearing B.1 lineage), Alpha (B.1.1.7), Beta (B.1.351), Gamma (P.1), Lambda (C.37 lineage) and Delta (B.1.617.2), Omicron BA.1 (B.1.1529.1), and Omicron BA.2 (B.1.1529.2) variant were prepared in our previous studies[15,28,29]. Plasmids expressing the point mutants were generated by site-directed overlap extension PCR using pC-SARS2-spike D614G or SARS2-Omicron-spike as the template and the following primers are listed in Supplementary Table 3. Primers for the construction of spike derivatives, related to Figs. 2 and 3. The resulting PCR fragment was digested with KpnI and NotI and inserted into the corresponding site of the pCAGGS vector. Nucleotide sequences were determined by Genetic Analyzer 3500xL (Applied Biosystems) and the sequence data were analyzed by GENETYX v12 (GENETYX Corporation).

## Intracellular cytokine staining

Intracellular cytokine staining was performed as previously described[13]. Briefly, A549-ACE2-A2402 cells ($3 \times 10^5$ cells) were transfected with 2 μg of plasmids expressing prototype spike or its derivatives using PEI Max (Polysciences, Cat# 24765-1) according to the manufacturer's protocol. At 2 days post transfection, the transfectants were harvested and mixed with the T-cell lines generated from HLA-A*24:02$^+$ vaccinated donors (see above) and incubated with RPMI 1640 medium (Thermo Fisher Scientific, Cat# 11875101) containing 10% FBS, 5 μg/ml brefeldin A (Sigma-Aldrich, Cat# B7651) in a 96-well U plate at 37 °C for 5 h. The cells were washed, and surface stained with following antibodies: CD3 FITC (UCHT1, 1/100 dilution), CD8 APCcy7 (RPA-T8, 1/100 dilution), CD14 PerCP/Cy5.5 (HCD14, 1/100 dilution), CD19 PerCP/Cy5.5 (HIB19, 1/100 dilution; Biolegend). Dead cells were stained with 7-aminoactinomycin D (Biolegend, Cat# 420404). After incubation at 4 °C for 20 min, the cells were fixed and permeabilized with a Cytofix/Cytoperm Fixation/Permeabilization solution kit (BD Biosciences, Cat# 554714) and were stained with IFN-γ PE (4 S.B3, 1/100 dilution; BD). After incubation at room temperature for 30 min, the cells were washed, and levels of protein expression were analyzed by flow cytometry using a FACS Canto II (BD Biosciences) followed by analysis using FACS Diva v9.0 (BD) and FlowJo v10 software (Tree Star).

## Western blotting

Transfected cells were lysed on ice for 15 min in a buffer (100 mM NaCl, 1 mM TCEP [Tris (2-carboxyethyl) phosphine hydrochloride], 2× protease inhibitor, and 10 mM HEPES; pH 7.5) containing 1% n-dodecyl-β-D-maltoside (DDM; Thermo Scientific). The resultant samples were resuspended in 1× Laemmli buffer containing 5% β-mercaptoethanol (Bio-Rad), boiled for 10 min, and subjected to protein separation by SDS-PAGE in 4−20% Mini-PROTEAN TGX precast gels (Bio-Rad) before transferred to nitrocellulose membranes (Wako). The membranes were incubated in a blocking buffer (Nacalai Tesque) for 1 h at room temperature and then mixed with primary antibodies, including rabbit anti-SARS-CoV-2 Spike (S1/S2) polyclonal antibody (1:2000; Invitrogen, Cat# PA5-112048) and mouse anti-β-actin monoclonal antibody (1:5,000; Wako, Cat# 010-27841), followed by staining with the horseradish peroxidase (HRP)-conjugated anti-rabbit (1:50,000; GE healthcare, Cat# NA934VS) and anti-mouse (1:25,000; GE healthcare, Cat# NA931VS) IgG secondary antibodies. The membrane was developed with the ImmunoStar LD enhanced chemiluminescence reagents (Wako) and visualized using ImageQuant LAS 4000 (GE Healthcare).

## Jurkat reporter cell (JurkatΔ-Luc) for functional analysis of TCRs

DNA fragment of NFAT-RE-Luc2P-SV40 pro-HygroR was amplified from pGL4.3 (Promega) by PCR. The DNA fragment was cloned into Stu I/Sal I site of PiggyBac vector PB530A-2 (SBI) by Gibson assembly method. The resultant vector [PB_NFAT-RE-Luc2P-SV40 pro-HygroR] was electroporated into endogenous TCR knocked-out Jurkat cells with Transposase expression vector PB200PA-1 (SBI). To select Jurkat reporter cell (JurkatΔ-Luc) integrated with NFAT-RE-Luc2P-SV40 pro-HygroR, Hygromycin-B selection was performed at 500 μg/ml concentration for 14 days.

## TCR cDNA amplification from single T cells and construction of TCR expression vector

The cryopreserved PBMCs were stained with NF9/A24 and QI9/A24 tetramers, anti-CD8 mAb (RPA-T8; Biolegend), and 7-amino-actinomycin D (7-AAD), and then tetramer$^+$CD8$^+$7-AAD$^-$ cells were sorted into 96-well plates (NIPPON Genetics, Cat# 4ti-0770/C) by using an FACS Aria II (BD Biosciences). TCRα and TCRβ cDNA pairs were amplified from single T cells by a one-step multiplex RT-PCR method described in our previous study[16]. The DNA sequences of the

PCR products were then analyzed by direct sequencing and the TCR repertoire by IMGT/V-QUEST (https://www.imgt.org/IMGT_vquest/vquest). The amplified TCRα and TCRβ cDNA fragments were connected to the missing constant region and linked to the blasticidin-S resistance (BlaR) gene by the Gibson assembly method with P2A ribosomal skipping sequences. Resultant TCRβ-P2A-TCRα-P2A-BlaR DNA was cloned into the PiggyBac vector (SBI, Cat# PB530A-2) by the Gibson assembly method.

## TCR-sensitivity assay

The plasmid PB TCR-P2A-BlaR was electroporated into JurkatΔ-Luc with Transposase vector (SBI, Cat# PB200PA-1) using Neon® Transfection System (Thermo Fisher Scientific) under the condition 1200 v, 5 ms, five pulses. After 48 h, JurkatΔ-Luc cells stably expressing TCRs were selected with RPMI medium containing 10 μg/ml of blasticidin-S for 10-14 days. These cells were cocultured with A549-ACE2-A2402 cells expressing each spike protein at an E:T ratio of 2:1 and incubated with RPMI 1640 medium (Thermo Fisher Scientific, Cat# 11875101) containing 10% FBS at 37 °C for 6 h. The mixture was measured for luciferase production using a luminescent substrate (Promega, Cat#E2510) by a CentroXS3 plate reader (Berthhold Technologies).

## Live virus suppression assay

A549 cells expressing ACE2/A2402 ($1 \times 10^4$ cells) were infected with each SARS-CoV-2 lineage at an MOI of 0.1 for 120 min at 37 °C. Cells were washed and cocultured with T cells at an E:T ratio of 2:1 and 1:1. Control wells containing virus-infected targets without T cells were also included. After 72 h incubation, culture supernatant was collected and performed real-time RT-PCR. 5 μl of culture supernatant was lysed in an equal amount buffer composed of 2% Triton X-100, 50 mM KCl, 100 mM Tris-HCl (pH 7.4), 40% glycerol and 0.4 U/μl recombinant RNase inhibitor (Promega, Cat# N2615) and then incubated at room temperature for 10 min. 90 μl of RNase-free water (Nacalai tesque, Cat# 06442-95) was added, and 3 μl of diluted sample was used as the template. Real-time RT-PCR analyses for viral RNA copy number was carried out with One-Step PrimeScript™ III RT-qPCR Mix (Takara, Cat# RR600B) and reactions were performed by using LightCycler® 96 System (Roche Diagnostics GmbH, Mannheim, Germany). For the primer, Primer/Probe N2 (2019-nCoV) (Takara, Cat# XD0008) were used as follows: NIID_2019-nCOV_N_ forward, 5'-AAATTTTGGGGACC AGGAAC-3'; NIID_2019-nCOV_N_ reverse, 5'-TGGCAGCTGTGTAGGTC AAC-3'; and NIID_2019-nCoV_N_ probe, 5'-FAM-ATGTCGCGCATT GGCATGGA-BHQ3. The viral RNA copy number was standardized with a Positive Control RNA Mix (2019-nCoV) (Takara, Cat#XA0142). Relative viral copy was calculated as viral RNA copy number obtained by virus-infected targets without T cells normalized to 1.

## Statistics and reproducibility

Data and statistical analysis were performed using Prism 9 (GraphPad Software). For two-way comparison, the paired Wilcoxon signed-rank test (Fig. 1e), unpaired Mann−Whitney $t$-test (Figs. 2d, 3e and 4b, c), or unpaired Student's $t$ test (Figs. 3c, d and 4a) was used. For multiple-group comparisons, one-way ANOVA with multiple comparisons by Bonferroni correction (Fig. 3f) was used.

In Figs. 2c, d, 3e, and Supplementary Fig. 1e, assays were performed in triplicate. In Figs. 3c, d, f; 4a−c; and Supplementary Fig. 2a, c, assays were performed in triplicate or quadruplicate. Data are representative of two or three independent experiments. In Supplementary Figs. 1g and 2b, representative blots of three independent experiments were shown.

## Reporting summary

Further information on research design is available in the Nature Research Reporting Summary linked to this article.

## Data availability

All data are present in this article and Supplementary Information files. The SARS-CoV-2 Wuhan strain data in this study have been deposited in the DDBJ (https://www.ddbj.nig.ac.jp/index.html) under accession ID LC528232. A B.1.617.2 (Delta), B.1.1.529 (Omicron/BA.1), and B.1.1.529 (Omicron/BA.2) lineage data have been deposited in the GISAID database (https://www.gisaid.org) under accession ID EPI_ISL_2080609, EPI_ISL_8559478, and EPI_ISL_9595859, respectively. The TCR data generated in this study are analyzed by the IMGT database (https://www.imgt.org/IMGT_vquest/vquest). Source data are provided with this paper.

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

## Acknowledgements

We would like to thank all members of the Genotype to Phenotype Japan (G2P-Japan) Consortium, especially Drs. Kei Sato (The University of Tokyo), Terumasa Ikeda (Kumamoto University) for providing plasmids, reagents, and cells, and Dr. Kazuhisa Yoshimura (Tokyo Metropolitan Institute of Public Health) and Drs. Tomohiko Takasaki and Jun-Ichi Sakuragi (Kanagawa Prefectural Institute of Public Health) for providing viruses. We thank Dr. Masafumi Takiguchi (Kumamoto University) for providing C1R-A2402 cells. We thank Dr. David L. Woodland for assistance in editing the manuscript. This study was supported in part by AMED Research Program on Emerging and Re-emerging Infectious Diseases 20fk0108539h0001 (to T.U.) and 20fk0108451s0101 (to T.U.) and AMED Research Program on HIV/AIDS 21fk0410046 (to C.M.), JSPS KAKENHI Grant-in-Aid for Scientific Research B 19H03703, 22H03119 (to T.U.) and 22H02877 (to C.M.), Scientific Research C 19K07623 (to C.M.) and 22K07089 (to M.T.), Takeda Science Foundation (to C.M. and M.T) and an intramural grant from Kumamoto University COVID-19 Research Projects (AMABIE) (to C.M.), IMAI MEMORIAL TRUST FOR AIDS RESEARCH (to M.T.), Shin-Nihon Foundation of Advanced Medical Research (to M.T.).

## Author contributions

C.M., M.T., T.S.T., Y.G., K.U. performed the experiments. Y.M., K.N., H.O., S.O., collected clinical samples. H.H., Y.A., M.K., H.K., prepared reagents. C.M., M.T., T.S.T., T.U., designed the experiments and interpreted the results. C.M., T.U., wrote the original manuscript. All authors reviewed and proofread the manuscript.

## Competing interests

The authors declare no competing interests.
