## [Peer Review File · Nature Communications]

The SARS-CoV-2 Omicron BA.1 spike G446S mutation potentiates antiviral T cell recognitionREVIEWER COMMENTS

Reviewer #1 (Remarks to the Author):

This paper describes a comprehensive analysis of T-cell responses to the SARS-CoV-2 Spike protein in recipients of mRNA Covid vaccines with the common Asian HLA class I allele, A*2402. The authors first demonstrate the immunodominance hierarchy of 3 A24-restricted epitopes in a large group of vaccinees, then look in depth at responses to the 2 immunodominant epitopes (NF9 and QI9) in different SARS-CoV-2 strains.

The main finding is that NF9 specific responses are enhanced towards spike proteins expressing the G446S substitution, found in Omicron BA.1, which lies outside the NF9 epitope, and this is reflected in the ability of some donor cell-lines to suppress viral replication of BA.1 better than other strains. Further characterisation demonstrates that the TCRs of NF9 specific cells are better able to respond to processed antigen, suggesting that this mutation affects antigen processing. Using a series of protease inhibitors, the authors conclude that the mutation enhances processing by the TPP11 protease (however, I note that whilst the use of the TPP11 inhibitor in figure 3f just reaches significance, the graph suggests that Bestatin also diminishes T-cell recognition).

Overall, this paper is well-written and clear, and the experiments have been carefully performed. I only have one comment:

In previous studies of viral (HIV) escape from antigen processing, it was shown that the mutations affected proteasomal processing. I do not understand why the authors did not look at proteasomal processing as well as enzymes involved in e.r. peptide trimming.

Reviewer #2 (Remarks to the Author):

In this manuscript Motozono and colleagues report a fascinating observation on a mutation within SARS-CoV-2 spike protein (G446S) which significantly enhances the endogenous presentation of a HLA A24-restricted epitope. Enhanced endogenous presentation is not due alteration the binding of the peptide epitope to HLA molecule or TCR recognition of MHC-peptide complex. Authors show that the enhanced recognition of NF9 epitope is due to improved endogenous processing through an ER resident tripeptidyl peptidase II. Interestingly, authors found that G446S mutation which is specifically seen in Omicron BA.1 variant and leads to enhanced immune recognition, loss of this mutation in BA.2 and delta variants does not enhance immune recognition. Overall this an elegantly designed study with very impressive data. Authors have conducted all experimental studies diligently and have provided strong supporting evidence to argue their case. I have few minor comments which authors may like to consider while revising their manuscript.

Data presented in Fig. 1d & e shows expression of CD25 and CD137 on T cell following stimulation with NF9 and QI9 peptide epitopes. I was bit surprised why authors did not use HLA-peptide tetramers for these epitope which they already used in the data presented Fig. 1C. Expression of these markers are highly unreliable as a marker for antigen specificity. If authors want to include this data, they should provide a proper controls. No peptide is not an appropriate control. I would suggest they use another viral peptide (e.g. HIV or influenza).

I was wondering if authors can provide pairwise analysis of T cell responses to NF9 and QI9 peptides in same donors. In addition, it would be nice if they can also include some data from individuals who have been infected with BA.1 and BA.2 variants to show the dynamics of T cell response to NF9 and QI9 peptides. Do authors have any clinical data from infected (symptomatic and asymptomatic) HLA A24+ individuals and how their T cell responses to NF9 and QI9 differ and evolve over the course of primary infection.

Please correct figure number in Line 215. This should read Fig. 3f not Fig. 3d.

Reviewer #3 (Remarks to the Author):

The authors provide a report of mutational changes in SARS-CoV-2 variants that have differential impact on T cell recognition. They demonstrate that while single amino acid changes in delta ablate recognition of an HLA-A24 restricted epitope the response is augmented in Omicron by an amino acid change that is adjacent to the epitope. They provide convincing evidence to support their hypothesis using cell lines over-expressing the spike variants and using infection with different viral variants.

Specific comments.

1. The manuscript would be enhanced if it was possible to provide real-world data on what happens to the magnitude of peptide restricted responses after exposure to Omicron or Delta.
2. Figure 1b: Tetramer staining isn't completely convincing given some background staining shown in A24- volunteers. Could be enhanced by included non-vaccinated, non-infected A24+ controls.
3. Figure 1c: How was the cut-off of 0.1% defined as a positive response

REVIEWER COMMENTS

**Reviewer #1 (Remarks to the Author):**

This paper describes a comprehensive analysis of T-cell responses to the SARS-
CoV-2 Spike protein in recipients of mRNA Covid vaccines with the common Asian
HLA class I allele, A*2402. The authors first demonstrate the immunodominance
hierarchy of 3 A24-restricted epitopes in a large group of vaccinees, then look in
depth at responses to the 2 immunodominant epitopes (NF9 and QI9) in different
SARS-CoV-2 strains.

The main finding is that NF9 specific responses are enhanced towards spike proteins
expressing the G446S substitution, found in Omicron BA.1, which lies outside the
NF9 epitope, and this is reflected in the ability of some donor cell-lines to suppress
viral replication of BA.1 better than other strains. Further characterisation
demonstrates that the TCRs of NF9 specific cells are better able to respond to
processed antigen, suggesting that this mutation affects antigen processing. Using
a series of protease inhibitors, the authors conclude that the mutation enhances
processing by the TPPII protease (however, I note that whilst the use of the TPPII
inhibitor in figure 3f just reaches significance, the graph suggests that Bestatin also
diminishes T-cell recognition).

Overall, this paper is well-written and clear, and the experiments have been carefully
performed. I only have one comment:

**Our reply:**

We appreciate Reviewer 1's positive comments and are happy to hear that
"*Overall, this paper is well-written and clear, and the experiments have been*
*carefully performed*".

As suggested by the reviewer, it could be interpreted as bestatin modestly reduced
the average of T-cell recognition (**Fig. 3f**), although the difference was not
statistically significant ($p = 0.2765$ by ANOVA, with multiple comparisons by
Bonferroni correction; versus DMSO alone). Accordingly, we have added the
sentence to mention this in the revised manuscript (**page 7, line 224-226**).

Additionally, we noticed that the statistical analysis of DMSO alone in **Fig. 3f** was
determined by ANOVA, with multiple comparisons by Bonferroni correction, but not
Mann-Whitney test in the original manuscript (**page 7, line 215 and page 15, line**

457-458). We sincerely apologize for the mistake. We have corrected the sentence
in the revised manuscript (page 7, line 223-224 and page 15, line 470-471).

In previous studies of viral (HIV) escape from antigen processing, it was shown that
the mutations affected proteasomal processing. I do not understand why the authors
did not look at proteasomal processing as well as enzymes involved in e.r. peptide
trimming.

**Our reply:**

Thank you very much for this important suggestion. We agree that this is a
possible scenario. To confirm whether G446S involves proteasomal processing,
we performed a TCR sensitivity assay in the presence of MG-132 (a proteasome
inhibitor). There was no statistical difference between the sensitivity of NF9/A24
and QI9/A24-specific TCRs in the presence of MG-132 ($p = 0.8850$ and >0.9999 by
unpaired two-tailed Student's t-test; versus DMSO alone, respectively). We
included this observation in the revised manuscript with the data (page 7, line 213-
218 and page 17, line 502-508; **Extended Data Fig. 2c**).

**Reviewer #2 (Remarks to the Author):**

In this manuscript Motozono and colleagues report a fascinating observation on a
mutation within SARS-CoV-2 spike protein (G446S) which significantly enhances the
endogenous presentation of a HLA A24-restricted epitope. Enhanced endogenous
presentation is not due alteration the binding of the peptide epitope to HLA molecule
or TCR recognition of MHC-peptide complex. Authors show that the enhanced
recognition of NF9 epitope is due to improved endogenous processing through an
ER resident tripeptidyl peptidase II. Interestingly, authors found that G446S mutation
which is specifically seen in Omicron BA.1 variant and leads to enhanced immune
recognition, loss of this mutation in BA.2 and delta variants does not enhance
immune recognition. Overall this an elegantly designed study with very impressive
data. Authors have conducted all experimental studies diligently and have provided
strong supporting evidence to argue their case. I have few minor comments which
authors may like to consider while revising their manuscript.

**Our reply:**

We are happy to hear that this reviewer feels that “*Overall this an elegantly designed*
*study with very impressive data*” and “*Authors have conducted all experimental*
*studies diligently and have provided strong supporting evidence to argue their case*”.

Data presented in Fig. 1d & e shows expression of CD25 and CD137 on T cell
following stimulation with NF9 and QI9 peptide epitopes. I was bit surprised why
authors did not use HLA-peptide tetramers for these epitope which they already used
in the data presented Fig. 1C. Expression of these markers are highly unreliable as
a marker for antigen specificity. If authors want to include this data, they should
provide a proper controls. No peptide is not an appropriate control. I would suggest
they use another viral peptide (e.g. HIV or influenza).

**Our reply:**

We thank the reviewer for this comment. The activation marker-induced (AIM) assay
has been extensively used to characterize antigen-specific T cell responses (Wolfl
et al. Blood, 2007. PMID: 17371945; Grifoni et al., Cell, 2020. PMID: 32473127;
Motozono et al. Cell Host Microbe, 2021. PMID: 34171266). We preliminarily
confirmed that there is no difference in the frequency of *in vitro*-expanded antigen-
specific T cells between AIM assay and tetramer staining, as shown below.

In addition to this assay (**Fig. 1d**), we confirmed the antigen-specificity of *in vitro*-
expanded T cell lines used for functional assay by HLA tetramer in **Extended data**
**Fig.1d**. However, we agree that the *in vitro* stimulation of PBMCs with the irrelevant
peptide would be better as a negative control. According to the reviewer's suggestion,
we stimulated PBMCs with representative HLA-A*24:02-restricted RF10 peptide
(RYPLTFGWCF, residues 134-143 of the HIV Nef protein) and included it in **the**
**revised Fig.1d and 1e** and manuscript (page 4, line 111, 112 and 114, page 14, line
432 and page 19-20, line 594-595). Dr. Yoshihiko Goto has been added as an author
to reflect the contribution of his experiments in the revised manuscript (page 1, line
5, 6, 15-17, 19, 21, 23, 25, and page 11, line 308).

I was wondering if authors can provide pairwise analysis of T cell responses to NF9
and QI9 peptides in same donors.

**Our reply:**

According to the reviewer's suggestion, we performed a pairwise analysis of T-cells
specific for the NF9 and the QI9 in the same donors in **the revised Fig.1c**. We have
added the data in non-vaccinated and seronegative HLA-A*24:02⁺ donors (n=5).
The new data are presented in **Fig. 1b and 1c** and (page 14, line 416-417, 419-
**422, and 430-431** of the revised manuscript).

In addition, it would be nice if they can also include some data from individuals who
have been infected with BA.1 and BA.2 variants to show the dynamics of T cell
response to NF9 and QI9 peptides. Do authors have any clinical data from infected
(symptomatic and asymptomatic) HLA A24⁺ individuals and how their T cell
responses to NF9 and QI9 differ and evolve over the course of primary infection.

**Our reply:**

We thank the reviewer for this interesting suggestion. In this study, we focused on
HLA-A*24:02-restricted vaccine-induced T cell responses against various SARS-
CoV-2 variants. However, we agree that it would be interesting to determine whether
an enhanced T cell response against Omicron BA.1 variant would be observed in
convalescents infected with Omicron BA.1 but not BA.2 variant, which is associated
with clinical outcome (severity, symptomatic or asymptomatic). We have started
collecting PBMC samples from convalescents with clinical data to address this
important question in a future study. Thank you again for the important suggestion.

Please correct figure number in Line 215. This should read Fig. 3f not Fig. 3d.

**Our reply:**

We sincerely apologize for the mistake. We have corrected it in the revised
manuscript (page 7, line 223).

**Reviewer #3 (Remarks to the Author):**

The authors provide a report of mutational changes in SARS-CoV-2 variants that
have differential impact on T cell recognition. They demonstrate that while single
amino acid changes in delta ablate recognition of an HLA-A24 restricted epitope the
response is augmented in Omicron by an amino acid change the is adjacent to the
epitope. They provide convincing evidence to support their hypothesis using cell
lines over-expressing the spike variants and using infection with different viral
variants.

**Our reply:**

We are happy to hear that this reviewer feels that “*They provide convincing evidence*
*to support their hypothesis using cell lines over-expressing the spike variants and*
*using infection with different viral variants*”.

Specific comments.

1.The manuscript would be enhanced if it was possible to provide real-world data on
what happens to the magnitude of peptide restricted responses after exposure to
Omicron or Delta.

**Our reply:**

We thank the reviewer for this important suggestion. We are now collecting SARS-
CoV-2-infected donors, rather than vaccine recipients in the current study, to
determine the magnitude of responses among peptides when exposed to Delta and
various sublineages of Omicron variants. We respectfully wish to address this
question in a future study.

2. Figure 1b: Tetramer staining isn't completely convincing given some background
staining shown in A24- volunteers. Could be enhanced by included non-vaccinated,
non-infected A24+ controls.

**Our reply:**

In accordance with the reviewer's comment, we performed additional experiments
using HLA-A*24:02+ non-vaccinated and seronegative donors (n=5). The new data
are presented in **Fig. 1b and 1c** and (page 14, line 416-417, 419-422, and 430-431
of the revised manuscript).

3. Figure 1c: How was the cut-off of 0.1% defined as a positive response

**Our reply:**

We used a cut-off value determined by the median plus 4 x SD in the negative
controls. To clarify this, we have added the sentence in the revised manuscript
(page 14, line 430-431).

REVIEWERS' COMMENTS

Reviewer #1 (Remarks to the Author):

The authors have responded to all my concerns and I am very happy with the revised paper.

Reviewer #2 (Remarks to the Author):

Authors have addressed all issues raised by me and have provided detailed rebuttal for each comments. Text and figures have been appropriately revised to address all comments. I don't have any further comments or concerns.

Reviewer #3 (Remarks to the Author):

The authors have adequately addressed the requests from the reviewers

**REVIEWER COMMENTS**

Reviewer #1 (Remarks to the Author):

The authors have responded to all my concerns and I am very happy with the revised
paper.

**Our reply:**

We appreciate Reviewer 1's comments and are happy to hear that "I am very happy
with the revised paper".

Reviewer #2 (Remarks to the Author):

Authors have addressed all issues raised by me and have provided detailed rebuttal
for each comments. Text and figures have been appropriately revised to address all
comments. I don't have any further comments or concerns.

**Our reply:**

We thank the reviewer for this positive reply and are happy to hear that 'Text and
figures have been appropriately revised to address all comments'.

Reviewer #3 (Remarks to the Author):

The authors have adequately addressed the requests from the reviewers

**Our reply:**

We sincerely thank the reviewer for this positive reply.
